

# Improved deep learning image classification algorithm based on Swin Transformer V2

Jiangshu Wei[1],[*], Jinrong Chen[1],[*], Yuchao Wang[2], Hao Luo[1] and Wujie Li[1]

[1] College of Information Engineering, Sichuan Agricultural University, Ya'an, Sichuan, China
[2] College of Mechanical and Electrical Engineering, Sichuan Agricultural University, Ya'an, Sichuan, China
[*] These authors contributed equally to this work.

## ABSTRACT

While convolutional operation effectively extracts local features, their limited receptive fields make it challenging to capture global dependencies. Transformer, on the other hand, excels at global modeling and effectively captures global dependencies. However, the self-attention mechanism used in Transformers lacks a local mechanism for information exchange within specific regions. This article attempts to leverage the strengths of both Transformers and convolutional neural networks (CNNs) to enhance the Swin Transformer V2 model. By incorporating both convolutional operation and self-attention mechanism, the enhanced model combines the local information-capturing capability of CNNs and the long-range dependency-capturing ability of Transformers. The improved model enhances the extraction of local information through the introduction of the Swin Transformer Stem, inverted residual feed-forward network, and Dual-Branch Downsampling structure. Subsequently, it models global dependencies using the improved self-attention mechanism. Additionally, downsampling is applied to the attention mechanism's Q and K to reduce computational and memory overhead. Under identical training conditions, the proposed method significantly improves classification accuracy on multiple image classification datasets, showcasing more robust generalization capabilities.

## INTRODUCTION

*Hubel & Wiesel (1960, 1962)* described the concept of neural receptive fields through their research on neurons in the visual cortex of cats, which laid the foundation for introducing the concept of convolutional kernels. In subsequent work, *LeCun et al. (1989)* and others proposed the core idea of convolutional neural networks. By employing convolutional kernels to perform convolutional operations on input images, they achieved the extraction and classification of image features. As a result, convolutional neural networks (CNNs) quickly found applications in the field of computer vision. The convolution operation, utilizing aggregation functions on local receptive fields (*Peng et al., 2021*), applies crucial

Corresponding author
Jiangshu Wei,
weijiangshu66@163.com

inductive biases during image processing, hierarchically collecting local features. The mechanism of shared convolutional kernel weights further facilitates the effective extraction of local features. Exploiting these advantages, CNNs have emerged as the mainstream framework in the field of computer vision, achieving state-of-the-art performance in tasks such as image classification (*Kim, Lee & Lee, 2016*), object detection (*Redmon et al., 2016*), and instance segmentation (*Zhu et al., 2021*). However, convolutional neural networks struggle to capture global dependencies due to the limited receptive field.

Recently, Transformer models based on the self-attention mechanism have become the mainstream framework for natural language processing (NLP) tasks. They have achieved great success in capturing long-range dependencies. Many studies have attempted to apply the self-attention mechanism to computer vision tasks, such as image classification, and have achieved promising results (*Wang et al., 2018*; *Bello et al., 2019*). Unlike previous approaches, the Vision Transformer (ViT) (*Dosovitskiy et al., 2020*) treated images as a sequence of tokens similar to word embeddings in NLP. It utilized a standard Transformer encoder to process the tokens, modeling the input image based on global relationships and extracting features. ViT has surpassed state-of-the-art performance on various image classification datasets and has dramatically inspired subsequent research (*Chen, Fan & Panda, 2021*; *Han et al., 2021*). However, although Transformers are proficient in modeling long-range dependencies within sequences, they lack a mechanism for local information exchange, which presents a disadvantage compared to traditional convolutional approaches. Firstly, since Transformers are designed initially for NLP tasks, the token dimensions within the Transformer blocks are one-dimensional, disregarding the 2D structure and local spatial information essential to images. Secondly, factors such as the quadratic complexity of token length, non-collapsible layer normalization, and GELU activation function contribute to frequent memory access (*Li et al., 2022b*), making the inference speed of Transformer models much slower than CNNs with the same number of parameters.

This article addresses the limitations of Transformers by leveraging the advantages of CNNs and proposes three technical improvements to enhance the performance of the Swin Transformer V2 model. The contributions of this article are as follows: Firstly, we replace the Patch Partition module and Linear Embedding module in the Swin Transformer V2 model with a Swin Transformer Stem. Three convolutional layers with different kernel sizes extract features from input images. This design aims to merge local features from different receptive fields in the generated feature maps, thus incorporating richer semantic information. Secondly, to decrease the model's computation and parameter load and make the model more lightweight, we adopt a Dual-Branch Downsampling structure to replace the Patch Merging module in the Swin Transformer V2 model. Simultaneously, by reducing the dimensionality of the feature maps, sensitivity to minor input details is reduced, thereby improving the model's generalization ability. Finally, to address the significant memory consumption and computational burden brought about by the attention module, we introduce average pooling layers within the basic blocks of the Swin Transformer V2. This is done to lower computational costs. To

maintain feature extraction effectiveness, a residual structure is introduced to mitigate the effects of downsampling. Furthermore, drawing from the inverted residual feedforward network structure of CMT model (*Guo et al., 2022a*), traditional convolutional layers and depth-wise separable convolutional layers are employed to enhance the extraction of local information.

## BACKGROUND

Since the introduction of AlexNet (*Krizhevsky, Sutskever & Hinton, 2017*), convolutional neural networks have emerged as the mainstream models for computer vision tasks (*Shepley et al., 2023*; *Wang et al., 2020*). VGGNet (*Simonyan & Zisserman, 2014*) confirmed the effectiveness of network depth in large-scale image classification CNNs, demonstrating that a convolutional neural network composed solely of convolutional and pooling layers can achieve state-of-the-art performance in image classification tasks. The residual learning function proposed by *He et al. (2016)* addressed the challenge of training deep neural networks. ResNet, which is eight times deeper than VGGNet, achieved higher accuracy while being easier to optimize and having lower complexity. The Inception structure proposed by *Szegedy et al. (2015)* and others indicated that significant improvements in recognition accuracy can be achieved by moderately increasing computational requirements. DenseNet (*Huang et al., 2017*) directly connected any two layers with the same feature map size, reducing computational complexity while improving recognition accuracy. Some research works have focused on reducing the number of model parameters and computational costs, such as ShuffleNet (*Zhang et al., 2018*), EfficientNet (*Tan & Le, 2019*), and MobileNets (*Mehta & Rastegari, 2021, 2022*), and in the context of transformers known for their excellent global modeling capabilities. *Guo et al. (2022b)* proposed VAN, which utilized large kernel convolutions to increase the receptive field, achieving global modeling with pure convolutional layers. ConvNeXt (*Liu et al., 2022b*), on the other hand, was a pure convolutional network model composed of standard convolutional network modules. It achieved performance comparable to advanced transformers, leading to a reevaluation of the importance of convolutions in computer vision. These outstanding convolutional neural network models have achieved remarkable results in image classification tasks. However, due to the limited receptive field of convolutional kernels in convolutional operations, despite the capability of these advanced models' convolutional layers to effectively extract local features, they have failed to capture global dependencies.

Following the tremendous success of Transformers in natural language processing tasks, many researchers have attempted to apply self-attention mechanisms and Transformers to computer vision tasks (*Pan et al., 2022*; *Shaker et al., 2023*). The groundbreaking model, Vision Transformer (ViT), directly employed Transformer encoders to process flattened one-dimensional image data, demonstrating the feasibility of a pure Transformer architecture in computer vision. To address the high complexity and computational requirements of the ViT architecture, Swin Transformer (*Liu et al., 2021*) introduced a hierarchical Transformer architecture that achieved linear computational complexity

relative to the image size while enabling multi-scale modeling. MViTs (*Fan et al., 2021*; *Li et al., 2022a*) performed pooling operations on the Q, K, and V components of the attention mechanism in Transformers to reduce the number of tokens, significantly reducing the computational overhead of the multi-scale Transformer architecture. Although Transformers excel at capturing long-range dependencies, they often overlook local feature details. Local details are crucial for visual tasks relating to lines, shapes, and object structures. To address this challenge, many researchers have attempted to incorporate convolutional neural networks (*d'Ascoli et al., 2021*; *Wang et al., 2022*) to complement the deficiency of Transformers in local feature extraction. CvT (*Wu et al., 2021*) improved the performance and efficiency of ViT by introducing a novel hierarchical structure that included convolutions and a convolutional Transformer block with convolutional projections. This work falls into the category of integrating the strengths of both CNNs and Transformers. It combines the local feature extraction capability of CNNs based on inductive biases with the global information capture capability of Transformers based on global modeling. The proposed approach enhances the performance of Swin Transformer V2 by integrating these two strengths.

# METHODS

## Model overall architecture

The improved model introduces three technical advancements to Swin Transformer V2. Firstly, the network stem of Swin Transformer V2 is replaced with the Swin Transformer Stem, replacing the original Patch Partition and Linear Embedding modules. Secondly, the Patch Merging module is replaced with a Dual-Branch Downsampling structure. Thirdly, convolutional networks and downsampling layers are introduced in the Swin Transformer blocks. Additionally, the self-attention mechanism in the Swin Transformer V2 blocks incorporates average pooling to reduce computational costs. The MLP module in the original Swin Transformer V2 is replaced with an inverted residual feed-forward network composed of convolutional layers to enhance the extraction of local features. The overall framework is illustrated in Fig. 1. The depicted model is the Swin Transformer V2-Tiny version, which inherits the original Swin Transformer V2 architecture. The input resolutions at each stage are downscaled by 4, 8, 16, and 32 factors, respectively.

In Swin Transformer V2, the input three-channel images were divided into patches of size $4 \times 4$. These patches were then flattened into one-dimensional vectors and linearly mapped to obtain tokens that can be input into the self-attention mechanism. However, it is evident that the divided patches do not preserve the 2D structural information of the image, and modeling the internal structure of the patches can only be achieved through poor linear projection. The present work introduces Swin Transformer Stem to address this limitation. This structure employs convolutional layers with different kernel sizes of 1, 3, and 5 to perform convolution operations with different receptive fields on the input image. The results from these operations are then concatenated to integrate local information across different channel directions. Subsequently, a $1 \times 1$ convolutional layer

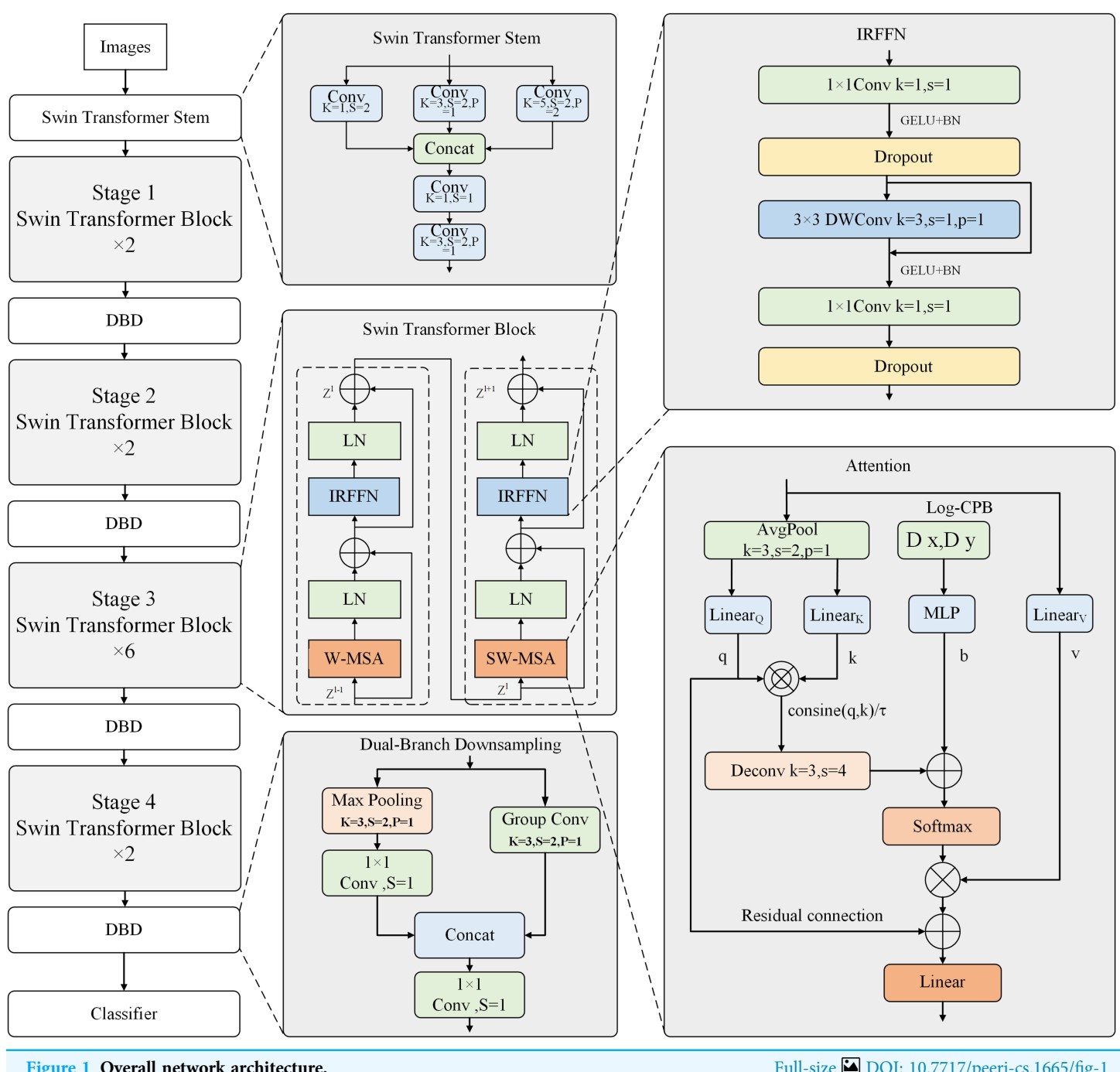

**Figure 1 Overall network architecture.**

is used to enhance the extraction of local information. Finally, a convolutional layer with a stride of 2 is employed to extract local features and achieve a 4× downsampling for the entire module, as shown in Fig. 2. The convolutional operations based on induction and bias enable better preservation of local information in the resulting feature map, which is lacking in the Transformer architecture.
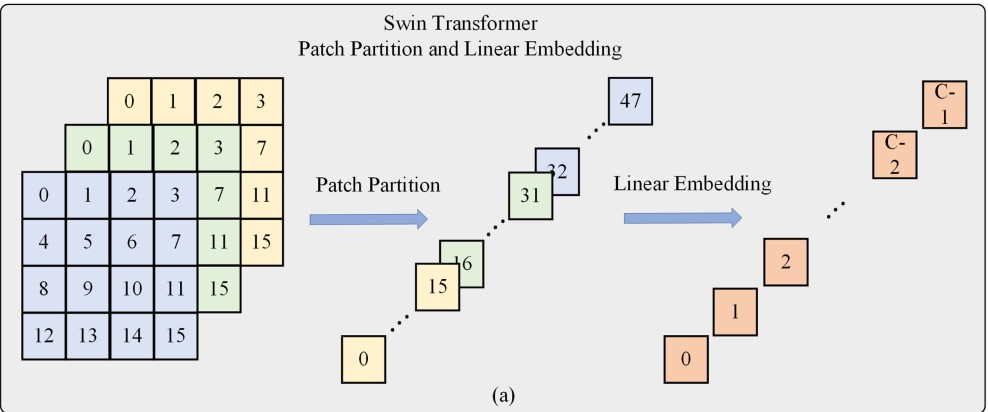
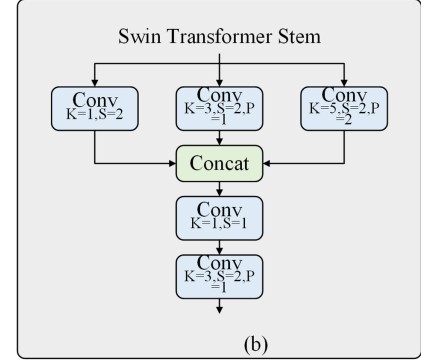

**Figure 2 Swin transformer stem.**

The Dual Branch Downsampling structure replaces the Patch Merging module in the original Swin Transformer V2 model. This design aims to downsample the input to the improved Swin Transformer blocks by a factor of 2 to reduce computational costs, as shown in the DBD module of the overall framework diagram. The first branch performs $3 \times 3$ max pooling to downsample the feature maps, then uses a convolutional layer to extract local information from the pooled features. The second branch employs a convolutional layer with a stride of 2 for local information extraction and downsampling by a factor of 2. The second branch utilizes grouped convolution to reduce computation. Finally, a $1 \times 1$ convolution is applied to fuse the features from the concatenated feature maps in the channel dimension.

## Improved attention mechanism

The traditional self-attention mechanism transforms the input X into query (Q), key (K), and value (V) vectors. In this mechanism, the similarity between pairs of elements is calculated as the dot product between the query vector Q and the key vector K (*Vaswani et al., 2017*), as shown in Eq. (1):

$$\text{Attn}(Q, K, V) = \text{Softmax}\left(\frac{QK^T}{\sqrt{d_k}}\right)V \tag{1}$$

However, the research conducted on Swin Transformer V2 (*Liu et al., 2022a*) has found that when this approach is applied to large-scale vision models, a few pixels often dominate the learned attention maps of specific blocks and heads. To address this issue, we adopt the scaled cosine attention method from Swin Transformer V2, which uses the logarithm of attention between pixel pairs i and j, scaled by a cosine function with natural normalization. This results in more gentle attention values, as shown in Eq. (2):

$$\text{Sim}(q_i, k_j) = \cos(q_i, k_j)/\tau + B_{ij} \tag{2}$$

**Peer**J Computer Science

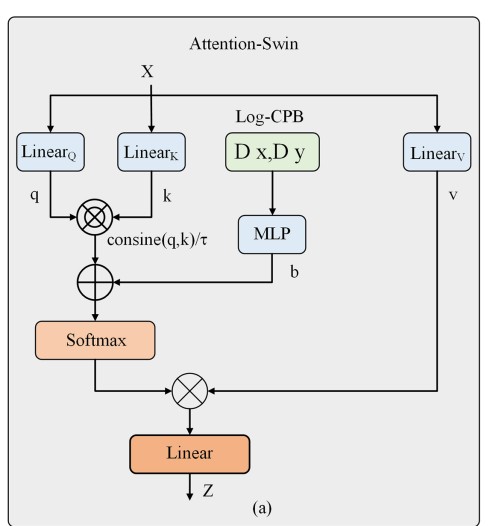
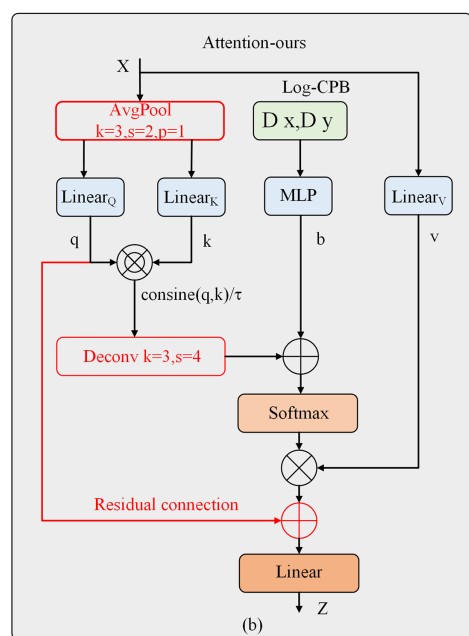

**Figure 3  Structural diagram of the improved attention mechanism.**

where $B_{ij}$ represents the relative positional bias between pixels i and j, and $\tau$ is a learnable scalar. The attention calculation is performed according to the following Eq. (3):

$$\mathrm{Attn}(Q, K, V) = \mathrm{Softmax}\left(\frac{Sim(QK^T)}{\sqrt{d_k}} + B\right) + V \tag{3}$$

However, the attention module of Swin Transformer V2, while capable of capturing global dependencies effectively, suffers from significant computational and memory overhead. Previous works have attempted to address this issue by simultaneously downsampling Q, K, and V or downsampling K and V. These downsampling operations were achieved using depth-wise separable convolutions. Inspired by these works, we propose a different approach in this article. Instead of using depth-wise separable convolutions, we introduce average pooling as a downsampling layer to reduce the computational complexity. Specifically, we perform 2× downsampling on Q and K using average pooling. Experimental results demonstrate that average pooling outperforms depth-wise separable convolutions for downsampling. Furthermore, motivated by the success of the residual link compensation pooling in MViTv2, we also introduce a residual link structure after the pooling operation on Q. As shown above, our improved attention mechanism differs from the original attention mechanism of Swin Transformer V2, as shown in Fig. 3. The computation of our improved attention mechanism is represented by the following Eq. (4):

$$\mathrm{Attn}(Q, K, V) = \mathrm{Softmax}\left(\frac{Sim(Q'K')}{\sqrt{d_k}} + B\right)V + Q \tag{4}$$

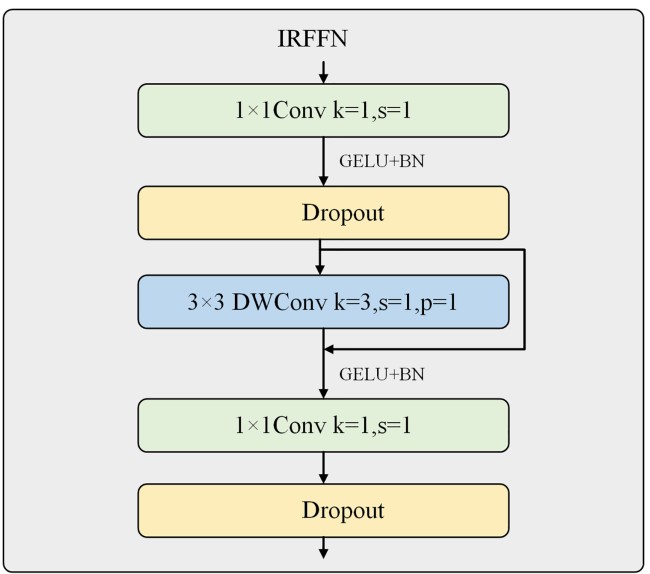

**Figure 4  Inverted residual feed-forward network.**

where $Q' = AvgPool(Q) \in R^{\frac{H \times W}{4} \times d_k}$, $K' = AvgPool(K) \in R^{\frac{H \times W}{4} \times d_k}$. The relative positional bias B is calculated based on logarithmic spatial coordinates. When computing attention, the relative positional bias is propagated between windows using an offset window. During this process, the extrapolation required is much smaller than what would be needed when using linear spatial coordinates.

## Inverted residual feed-forward network

The inverted residual feed-forward network is an essential compensation for the Swin Transformer V2's limited ability to extract local features. It is designed to be placed at the end of the Swin Transformer Block to extract more important and prominent features, as shown in Fig. 4.

The MLPs used in the ViT and Swin Transformer V2 consist of two fully connected and two Dropout layers. The first fully connected layer has four times the number of neurons as the input channels and is activated using the GELU activation function. The second fully connected layer restores the original number of channels. In contrast, CMT and NextViT attempted to replace the fully connected layers with convolutional layers to extract more local features. This work introduces the CMT network's IRFFN structure, consisting of two $1 \times 1$ convolutional layers, two Dropout layers, and a depth-wise separable convolutional layer. The two convolutional layers expand the number of channels by four times and then restore it to the original number of input channels, allowing for further extraction of local features. The convolutional layer uses a $3 \times 3$ kernel size and incorporates residual structures to improve gradient propagation. The use of depth-wise separable convolution ensures that the computational cost is negligible.

**Table 1 Dataset characteristics.**

| Datasets | Classes | Total number | Training set number | Testing set number |
|---|---|---|---|---|
| The Oxford-IIIT pet | 37 | 7,390 | 5,913 | 1,477 |
| Imagewoof | 10 | 12,954 | 9,025 | 3,929 |
| 102 Category flower | 102 | 8,189 | 6,587 | 1,602 |

## RESULTS

### Experiment datasets

To validate the improvement of the enhanced model's performance, this study conducts experiments on three datasets with significantly different numbers of classes. These datasets include the Oxford 102 Category Flower Dataset (*Nilsback & Zisserman, 2008*), the Oxford-IIIT Pet Dataset (*Parkhi et al., 2012*), and the Imagewoof Dataset (*Howard & Gugger, 2020*) from the University of San Francisco. The Oxford-IIIT Pet Dataset consists of a collection of pet images with 37 different categories, exhibiting significant scale, pose, and lighting variations. The Imagewoof dataset is a subset of 10 categories from the Imagenet dataset, explicitly focusing on dog breeds, which presents a challenging classification task due to the similarity among the categories. The dog breeds included in this dataset are Australian terrier, Border terrier, Samoyed, Beagle, Shih Tzu, English foxhound, Rhodesian ridgeback, Dingo, Golden retriever, and Old English sheepdog. The 102 Category Flower Dataset contains images of 102 different flower categories, showcasing significant scale, pose, and lighting variations. This dataset includes categories with significant intra-class variations and several closely related categories. The characteristics of these three datasets are summarized in Table 1.

### Model evaluation metrics

This article employs three evaluation metrics, namely computational cost, Top-1 accuracy, and Top-5 accuracy, to assess the improved model. Computational cost refers to the total computational operations executed during training or inference. It is commonly used to evaluate the computational complexity of a model, aiding in measuring its performance on different hardware devices and its computational resource requirements. In this study, we utilize the ptflops toolkit from the PyTorch scientific computing library to quantify the computational cost of the enhanced model. Accuracy, on the other hand, signifies the proportion of correctly predicted samples by the classification model out of the total number of samples, serving as a measure of the model's performance in image classification tasks. This is a common performance evaluation metric, as illustrated in Eq. (5):

$$\text{Accuracy} = \frac{TP + TN}{TP + FN + FP + TN} \tag{5}$$

where TP represents the number of instances that are labeled as positive and correctly classified as positive, FN stands for the number of instances labeled as positive but mistakenly classified as negative, FP signifies the number of instances labeled as negative

but incorrectly classified as positive, and TN corresponds to the number of instances labeled as negative and correctly classified as negative. When the model predicts an image, it provides N probabilities for each class, indicating the network's predictions of the likelihood that the test image belongs to each class. Top-1 accuracy refers to the accuracy of the class that ranks first among the N (where N > 1) class probabilities, matching the actual class label, as depicted in Eq. (6):

$$\text{Accuracy} = \frac{\text{First}^{(TP+TN)}}{TP + FN + FP + TN} \tag{6}$$

where $\text{First}^{()}$ denotes the count of instances where the highest classification probability matches the label, similar to Top-1 accuracy, Top-5 accuracy refers to the accuracy of the top five ranked classes among the N (where N > 5) class probabilities, which includes the actual result, as shown in Eq. (7):

$$\text{Accuracy} = \frac{\text{Five}^{(TP+TN)}}{TP + FN + FP + TN} \tag{7}$$

where $\text{Five}^{()}$ denotes the count of instances where the correct label is among the top five highest classification probabilities. A more minor computational cost indicates lower resource requirements for deploying the model among the three evaluation metrics. Conversely, higher Top-1 and Top-5 accuracies signify the model's more robust classification capability.

## Comparison with previous models

The improved network architecture selected for this experiment is the Swin Transformer V2 Tiny version. The input consists of three-channel images with a size of $256 \times 256$ pixels. The window size used for partitioning the images and computing attention is $16 \times 16$. In the first stage, the number of channels is set to 96. The four stages of the network consist of repeated blocks with the following repetition counts: 2, 2, 6, and 2.

In this experiment, the network is trained using a batch size of 64 for the input images. The training is performed for 300 epochs. Data augmentation is applied using the mixup technique. The activation function used is AdamW optimizer with weight decay. The learning rate is decayed using the cosine annealing method with the warmup. To ensure fairness in the experiment, no pre-training is used, and the training settings are kept identical. The highest accuracy achieved within 300 iterations is considered the absolute accuracy. Detailed training parameters are provided in Table 2. For comparison, the models used include ViT, PVTv2, Swinv1, CMT, MViTv2, and Swinv2. All experiments are conducted under the same conditions. The model parameters, computational cost, and accuracy of the three datasets are shown in Table 3.

In order to verify the improvement in accuracy of the enhanced model relative to the excellent model, this study conducts experiments using the PVT, MViT, CMT, and Swin Transformer models. Among these, the PVT model is based on the ViT model and introduces a pyramid structure, creating a pure Transformer model. The MViT model, building upon a pure Transformer architecture, incorporates pooling operations to reduce

**Table 2 Parameterization of model and training.**

| Model parameter | Value | Training parameter | Value | Training parameter | Value |
|---|---|---|---|---|---|
| Image size | 256 | Batch size | 64 | Warmup lr | 0.0000005 |
| Windows size | 16 | Epochs | 300 | Scheduler | Cosine |
| Embed dim | 96 | Warmup epochs | 20 | Decay epochs | 30 |
| Depth | 2,2,6,2 | Weight decay | 0.05 | Decay rate | 0.1 |
| Num heads | 3,6,12,24 | Optimizer | AdamW | Random erase mode | Pixel |
| Drop path rate | 0.2 | Base lr | 0.0005 | Mixup alpha | 0.8 |

**Table 3 Comparison of network performance on three datasets.**

| Model | Flops (G) | Imagewoof | | The Oxford-IIIT pet | | 102 Category flower | |
|---|---|---|---|---|---|---|---|
| | | Top-1 Acc | Top-5 Acc | Top-1 Acc | Top-5 Acc | Top-1 Acc | Top-5 Acc |
| PVTv2_B1 | 2.1 | 85.4% | 97.5% | 79.7% | 93.5% | **95.6%** | 98.7% |
| Swinv1_Tiny | 4.5 | 78.8% | 97.7% | 70.7% | 93.7% | 94.1% | 98.4% |
| CMT_S | 4.0 | 79.7% | 96.7% | 59.0% | 82.3% | 74.9% | 87.5% |
| MViTv2_Tiny | 4.7 | 85.6% | 97.2% | **83.2%** | 96.0% | 95.3% | 98.9% |
| Swinv2_Tiny | 6.6 | 75.8% | 97.6% | 66.4% | 91.2% | 91.1% | 97.6% |
| **Ours** | 5.7 | **87.6%** | **98.7%** | 82.6% | **96.8%** | 95.4% | **98.8%** |

Note:
The results of the model presented in this article are shown in bold.

the computational load. In contrast, the CMT model merges convolutional neural networks into the Transformer architecture to enhance the extraction of local features, achieving a fusion of Transformer and CNN capabilities. By comparing the accuracy results from the experiments, it is evident that the improved model in this research outperforms both pure Transformer network models and models incorporating CNN layers in terms of performance. This validates the effectiveness of the improved model, which leverages the strengths of both Transformer and CNN. Compared to the original Swinv2 Tiny model, the improved model shows significant improvements on The Oxford-IIIT Pet, Imagewoof, and 102 Category Flower datasets, with an increase of 16.2%, 11.8%, and 4.3%, respectively. This demonstrates its more robust generalization capability. The improved model also achieves higher accuracy compared to the pure Transformer-based model, the PVT and MViT models based on improved Transformers, and the CMT model with introduced CNN layers.

## Ablation experiments result

In this section, ablation experiments are conducted to validate the effectiveness of the proposed components: Swin Transformer Stem, Dual-Branch Downsampling, and the improved Swin Transformer Block. The experimental results are presented in Table 4.

It can be observed that the improved model demonstrates more robust performance, with an increase of 16.18, 11.83, and 4.3 percentage points on the three datasets, respectively. The introduction of convolutional layers to compensate for the limitations of

**Table 4 Performance of the improved model on three datasets.**

| | Block | | DBD | Stem | Accuracy(ours) |
|---|---|---|---|---|---|
| | DWConv | AvgPool | | | |
| The Oxford-IIIT pet | √ | × | × | × | 66.96(+0.54)% |
| | √ | × | √ | × | 76.98(+10.56)% |
| | × | √ | √ | × | 77.05(+10.63)% |
| | × | √ | √ | √ | 82.60(+16.18)% |
| Imagewoof | √ | × | × | × | 79.99(+4.19)% |
| | √ | × | √ | × | 86.21(+10.41)% |
| | × | √ | √ | × | 86.69(+10.89)% |
| | × | √ | √ | √ | 87.63(+11.83)% |
| 102 Category flower | √ | × | × | × | 93.57(+2.43)% |
| | √ | × | √ | × | 94.19(+3.05)% |
| | × | √ | √ | × | 94.32(+3.18)% |
| | × | √ | √ | √ | 95.44(+4.30)% |

the Transformer in capturing local features resulted in accuracy improvements on all three datasets. Specifically, there is an increase of 4.19 percentage points on the Imagewoof dataset, 2.43 percentage points on the 102 Category Flower dataset, and 0.54 percentage points on The Oxford-IIIT Pet dataset. Comparing the results before and after incorporating the Dual-Branch Downsampling structure, it can be observed that the model's generalization ability improved when combining the feature maps extracted by the convolutional layer with the downsampling layer that retains translational invariance. The improvements are 10.56 percentage points on The Oxford-IIIT Pet dataset, 10.41 percentage points on the Imagewoof dataset, and 3.05 percentage points on the 102 Category Flower dataset. The experiment also attempted to extract local features once again using depth-wise separable convolutional layers in the downsampling stage of the attention mechanism. However, the data indicates that using average pooling yielded better results, with a 0.48 percentage point improvement on the Imagewoof dataset. The model's accuracy is further enhanced after applying the Swin Transformer Stem, which extracts local information with different receptive fields and fuses feature maps on the channel dimension. The improvements on the three datasets reach 16.18, 11.83, and 4.3 percentage points, respectively.

It can be observed that the final accuracy of the proposed improved model is higher than that of both the advanced pure Transformer model and the model combining convolutional neural networks, which greatly depends on several network modules proposed in this study. Firstly, the convolutional layers in Swin Transformer Stem utilize convolutional kernels of different sizes, enabling the extraction of local features at different scales and preserving more semantic information. Secondly, the concatenation operation is employed for feature map fusion in the downsampling structure of the dual branch, which also helps retain rich semantic information along the channel dimension. Additionally, the

improved Block enables global modeling and further extracts local features through IRFFN. Although this network model performs well, it still has some limitations. For instance, compared to lightweight pure convolutional neural networks, the improved model has higher accuracy but more significant parameter and computational requirements, resulting in slower inference speeds. Therefore, it not easy to deploy to mobile or industrial scenarios at this time.

## CONCLUSION

In this work, we conduct technical enhancements to the Swin Transformer V2 model, resulting in an improved version that combines the strengths of both the Transformer and CNN. In this enhanced model, we integrate the advantages of Transformer and CNN by introducing the Swin Transformer Stem, a dual-branch downsampling structure, and a reverse feedforward network. These modifications endow the model with the ability to extract local information while capturing long-range dependencies effectively. Furthermore, we apply average pooling to the self-attention mechanism, thereby reducing the computational load of the model. Experimental results demonstrate that when compared to the Swin Transformer V2-Tiny network, the improved model effectively leverages the strengths of both CNN and Transformer, resulting in significant performance improvements.

In future research endeavors aimed at further enhancing model accuracy and exploring the potential of image classification, we intend to undertake the following measures: We plan to adjust the number of blocks in each stage to improve model accuracy. By appropriately increasing or decreasing the number of blocks, we can optimize the balance between model complexity and performance. Secondly, we intend to design more suitable feature fusion strategies. In the first and second stages, we plan to forgo self-attention mechanisms in favor of lighter-weight convolutional modules. This approach can capture different scale and fine-grained feature information while reducing computational overhead. Additionally, given the subpar classification performance of the improved model on large datasets, we plan to introduce a feature pyramid structure. This will empower different-scale features in each stage with solid semantic information. Consequently, the model will be able to fuse robust semantic information from low-resolution feature maps and rich spatial information from high-resolution feature maps simultaneously.

## ACKNOWLEDGEMENTS

We would like to thank Professor Li for his helpful comments on the article.

### Funding

This work was supported by the University-Industry Collaborative Education Program (Grant NO. 22097077265201). The funders had no role in study design, data collection and analysis, decision to publish, or preparation of the manuscript.

## Grant Disclosures

The following grant information was disclosed by the authors:
University-Industry Collaborative Education Program: 22097077265201.

## Competing Interests

The authors declare that they have no competing interests.

## Author Contributions

- Jiangshu Wei conceived and designed the experiments, authored or reviewed drafts of the article, and approved the final draft.
- Jinrong Chen conceived and designed the experiments, performed the experiments, analyzed the data, performed the computation work, prepared figures and/or tables, authored or reviewed drafts of the article, and approved the final draft.
- Yuchao Wang analyzed the data, performed the computation work, authored or reviewed drafts of the article, and approved the final draft.
- Hao Luo performed the experiments, performed the computation work, prepared figures and/or tables, and approved the final draft.
- Wujie Li analyzed the data, prepared figures and/or tables, and approved the final draft.

## Data Availability

Improved Swin Transformer v2 code: Jinrong Chen. (2023). Code of Improved Swin Transformerv2. https://doi.org/10.5281/zenodo.8068300.

The Oxford-IIIT Pet dataset link:

https://www.robots.ox.ac.uk/~vgg/data/pets/.

Imagewoof dataset link: https://github.com/fastai/imagenette.

102 Category Flower dataset link: Jinrong Chen. (2023). The 102 Category Flower Dataset [Data set]. https://www.robots.ox.ac.uk/~vgg/data/flowers/102/categories.html.

## Supplemental Information

Supplemental information for this article can be found online at http://dx.doi.org/10.7717/peerj-cs.1665#supplemental-information.

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
