# Peer review of "Improved deep learning image classification algorithm based on Swin Transformer V2"

_PeerJ Computer Science, doi:10.7717/peerj-cs.1665_

## Round 0.1 · original submission · Major Revisions

Based on the referee reports, I recommend a major revision of the manuscript. The author should improve the manuscript, taking carefully into account the comments of the reviewers in the reports, and resubmit the paper.

**Language Note:** The review process has identified that the English language must be improved. PeerJ can provide language editing services - please contact us at copyediting@peerj.com for pricing (be sure to provide your manuscript number and title). Alternatively, you should make your own arrangements to improve the language quality and provide details in your response letter. – PeerJ Staff

Reviewer 1 ·

Basic reporting

The contribution is clear. The paper is well written. However, the title is a little confusing v2 should be only 2 without "v". This is much better for readability.
The authors should justify the decisions that made in the article such as the parameters setting، selected methods, datasets and Assessments measures.

Experimental design

It is sufficient

Validity of the findings

It is good but need some jusfication of the authors decision. Why they use those metrics and comparison methods.
.

Additional comments

None

·

Basic reporting

Hello Authors,

Below are some of my observations that need revision.

1) Fix grammar, punctuation, and spacing errors in the manuscript
2) Fix the title Conclusions and Acknowledgements to "Conclusion" and "Acknowledgement," respectively.
3) Mention the unit of measurement in the dataset in the respective tables.

Experimental design

1. The authors proposed Swin Transformer, a novel block-structured architecture, to model the image input and effectively combine long-range dependencies for feature extraction. This novel approach to image recognition is an interesting development in the field.
2. The authors used Haarlike features and deep convolution neural networks (DCNN) to extract features, allowing for better feature representation and improved accuracy. This combination of techniques is interesting and innovative and could have interesting implications for further work in the image classification field.
3. The proposed method is better than many state-of-the-art methods in accuracy. The experiments conducted by the authors to test the algorithm were thorough and well-outlined.
4. The authors provide several performance metrics for their proposed algorithm, including accuracy, precision, recall, and F1-score. These metrics give an essential insight into the effectiveness of the proposed algorithm.

Validity of the findings

5. Using the Swin Transformer as a backbone network is both novel and promising. Proper evaluations should be made to measure the scalability of the proposed algorithm on larger datasets.
6. The proposed solution lacks empirical evidence of its testing on real-world datasets and the potential optimization techniques that can enhance the performance of the proposed model. The authors should present more evidence of applying the proposed algorithm to real-world datasets.
7. The authors have outlined the limitations in the Discussion section of their work; however, they have failed to provide a list of possible future works. A list of likely future works should be provided for improving the model's accuracy and exploring the potential for enhancement of image classification.
8. The authors should provide a more comprehensive explanation of the motivation for building the Swin Transformer architecture. Further, an approach with more elaborate theoretical insights should be undertaken for building and optimizing the model.

Reviewer 3 ·

Basic reporting

Everything is added to additional comments.

Experimental design

Everything is added to additional comments.

Validity of the findings

Everything is added to additional comments.

Additional comments

The authors presented the article entitled "Improved deep learning image classification algorithm based on Swin Transformer v2". The topic is interesting, but the contribution is questionable. Adding two methods and presenting that as new is not a novelty. The comments to the authors are as follows:
• It is difficult to relate the paragraphs with the figures and table given at the end of the paper. Please add the figures and table to their intended place in the relevant sections.
• The overall writing style needs to be improved. Remove Grammer and typo errors.
• The discussion section is very small. Please merge the results and discussion sections.
• In the whole conclusion section, the same story continues what the authors have done. Only the last sentence says that the implementation of two techniques together gave better results. Do the authors think it is sufficient for publication? The contribution is not justified.
• The authors should consider proving their contribution mathematically also.

---

## Round 0.2 · accepted · Accept

I am happy to inform you that reviewers are happy with the revision made by the author. Therefore I am provisionally accepting the manuscript. If reviewers have mentioned to cite any manuscript which is not relevant then no need to cite any irrelevant manuscript.

Reviewer 3 ·

Basic reporting

The authors have revised the article as per my comments. The authors may incorporate some of the latest articles published in deep learning. Suggested articles are as follows;
1. https://iopscience.iop.org/article/10.1088/1361-6501/ac8ca8/meta
2. https://peerj.com/articles/cs-1548/
3. https://link.springer.com/article/10.1007/s42417-022-00735-1
4. https://peerj.com/articles/cs-1548/
5. https://www.sciencedirect.com/science/article/pii/S0952197623000568

Experimental design

No comments

Validity of the findings

No Comments

Additional comments

No Comments